# Impact of a culturally tailored parenting programme on the mental health of Somali parents and children living in Sweden: a longitudinal cohort study

Fatumo Osman  ,[1,2] Linda Vixner,[1] Renee Flacking,[1] Marie Klingberg-Allvin,[1] Ulla-Karin Schön,[3] Raziye Salari[2]

¹School of Education, Health and Social Studies, Hogskolan Dalarna, Falun, Sweden
²Department of Public Health and Caring Sciences, Uppsala University, Uppsala, Sweden
³Department of Social Work, Stockholm University, Stockholm, Sweden

**Correspondence to**
Fatumo Osman; fos@du.se

## ABSTRACT

**Objectives** This study aimed to evaluate the long-term impact (3-year follow-up) of a culturally tailored parenting support programme (Ladnaan) on the mental health of Somali-born parents and their children living in Sweden.

**Methods** In this longitudinal cohort study, Somali-born parents with children aged 11–16 were followed up 3 years after they had participated in the Ladnaan intervention. The Ladnaan intervention comprises two main components: societal information and the Connect parenting programme delivered using a culturally sensitive approach. It consists of 12 weekly group-based sessions each lasting 1–2 hours. The primary outcome was improved mental health in children, as measured by the Child Behaviour Checklist (CBCL). The secondary outcome was improved mental health in parents, as measured by the General Health Questionnaire-12. Data were collected from the parent's perspective.

**Results** Of the 60 parents who were originally offered the intervention, 51 were included in this long-term follow-up. The one-way repeated measures (baseline to the 3-year follow-up) analysis of variance for the CBCL confirmed maintenance of all the treatment gains for children: total problem scores (95% CI 11.49 to 18.00, d=1.57), and externalising problems (95% CI 2.48 to 5.83, d=0.86). Similar results were observed for the parents' mental health (95% CI 0.40 to 3.11, d=0.46).

**Conclusion** Positive changes in the mental health of Somali-born parents and their children were maintained 3 years after they had participated in a parenting support programme that was culturally tailored and specifically designed to address their needs. Our findings highlight the long-term potential benefits of these programmes in tackling mental health issues in immigrant families.

**Trial registration number** NCT02114593.

## STRENGTHS AND LIMITATIONS OF THIS STUDY

⇒ The research used a longitudinal cohort study design (3-year follow-up) of an intervention group who received support from a culturally tailored parenting programme.
⇒ The majority of the original sample were retained in this 3-year follow-up.
⇒ The main limitation of the study was the lack of a control group, which only allowed within-group analyses.
⇒ The control group was offered the intervention shortly after the 2-month follow-up due to ethical and practical reasons.
⇒ Data were collected through parental self-report.

## INTRODUCTION

Over the past 20 years, Sweden and many other European countries have experienced a rapid increase in immigration, particularly immigrants who have fled from areas of war.[1] Stress factors due to immigration have a negative influence on the mental health of parents and their children.[2–7] More specifically, studies have shown that postmigration factors such as lack of social networks and support, perceived discrimination in the host county, inadequate accommodation and acculturation challenges contribute to behavioural problems in children.[7–13] Studies have also reported that the mental health problems of immigrant parents are associated with low confidence in parenting[14–16] and the use of authoritarian parenting, which has been shown to be related to children externalising problems.[6 17 18] In addition to negative parenting practices, power conflicts between parents and children are also associated with more behavioural problems among immigrant children.[17 18]

There is a large body of evidence indicating that parenting support programmes promote positive parenting practices and parent–child relationships, decrease children's internalising and externalising problems, and improve parents' mental health problems.[19 20] However, parents with an immigrant background have reported a need for culturally tailored parenting support programmes that assist them in adapting their parenting to the host country, improving their relationships

with their children,[7 21–24] and improving their children's mental health.[25] Few studies have attempted to culturally tailor parenting programmes to immigrant families. These studies have reported that culturally tailored parenting programmes improve children's mental health, strengthen parenting roles and enhance parent–child relationships in the short term.[26–29] However, little is known about the long-term effects of these programmes on the mental health of parents and their children.

## Objectives

This study aimed to evaluate the long-term impact (3 years follow-up) of a culturally tailored parenting support programme (Ladnaan) on the mental health of Somali-born parents and their children living in Sweden.

## Methods study design

This cohort study is a longitudinal (3 years) follow-up of the intervention group in a randomised controlled trial,[27 28] which evaluated the impact of a culturally tailored parenting support programme for Somali-born parents living in Sweden, referred to as the Ladnaan intervention.

Originally, the study included a 2-month and a 6-month postintervention follow-up for both intervention and control groups. However, the participating municipality started offering the intervention to the control group shortly after the 2-month follow-up data collection. As we had lost most of our control group by the sixth month, we opted for a 3-year postintervention follow-up instead.

## Setting and participants

The study was conducted in a town in the middle of Sweden with approximately 50 000 inhabitants, of whom approximately 3000 are of Somali origin. Baseline (preintervention) and 2-month follow-up data were collected between May 2014 and October 2015. The 3-year postintervention data were collected between April and October 2018. Data were collected by the first author and by research assistants who were trained in the recruitment process, taking informed consent and the use of the instruments employed in the study.

The main trial included 120 Somali-born parents with children aged 11–16 years and self-perceived stress related to parenting practices. If the parents had more than one child aged 11–16, they were asked to complete the questionnaire for the child with whom they particularly wished to improve their relationship. In two-parent families, both parents were offered the possibility of participating in the intervention, but, throughout the study, only the parent who was screened at baseline completed the questionnaire. Originally, we had also planned to collect data from the children. However, the parents did not consent to their children's direct participation in the study (mainly due to the fear of Social Services). Thus, only parent-reported data were available.

Baseline data were collected prior to randomisation and the follow-up data were collected 2 months as well as

3 years postintervention. In comparison to the parents in the control group, the parents in the intervention group reported significant improvement in their children's and their own mental health at the 2-month follow-up.[27 28] Since most of the parents in the control group received the intervention soon after the 2-month follow-up, the present paper focuses on the 3-year follow-up of the original intervention group. Of the 60 parents in the intervention group who completed the baseline assessment, three were lost to the 2-month follow-up, and six were lost to the 3-year follow-up. Thus, 51 parents in the intervention group were followed up 3 years postintervention (see figure 1).

## Patient and public involvement statement

A reference group consisting of key persons from the Somali community, Somali associations, Social Services, School Health and other stakeholder organisations were engaged in our original study. The reference group has given feedback on the initial design, conducting the project and disseminating the results.

## Intervention

The Ladnaan intervention, developed based on a qualitative study conducted prior to the trial,[7] consists of two main components: societal information and the Connect parenting programme.[30] It included 12 weekly sessions with groups of 12–17 parents, each lasting approximately 1–2 hours.

The first two sessions covered societal information relevant to Somali-born parents, including an overall view of the Swedish Child Welfare Services, the Convention on the Rights of the Child and parenting styles in Sweden. The societal information component was developed through a qualitative study conducted with Somali-born parents that aimed to explore Somali-born parents' need for parenting support.[7] The other 10 sessions covered the Connect parenting support programme, which is a standardised programme based on attachment theory. The Connect programme addresses nine different principles of child development, parent–child relationships and challenging interactions, and included role playing, case examples and reflection exercises.[30] (Detailed information of each session is shown in online supplemental table 1). The cultural sensitivity delivery approach was employed to ensure the sustainability of the parenting programme after the study was finalised. This delivery approach incorporated elements, such as delivering the programme in the participants' native language, using group leaders of similar background and experience, and culturally tailoring the role playing and case examples to make them understandable and relatable for the participants. Only the parenting style theme in the societal information sessions was delivered by a Swedish-speaking professional from the Family and Child Welfare Services and interpreted by one of the group leaders.

In connection with every session, the parents were offered support for reading and writing letters from the

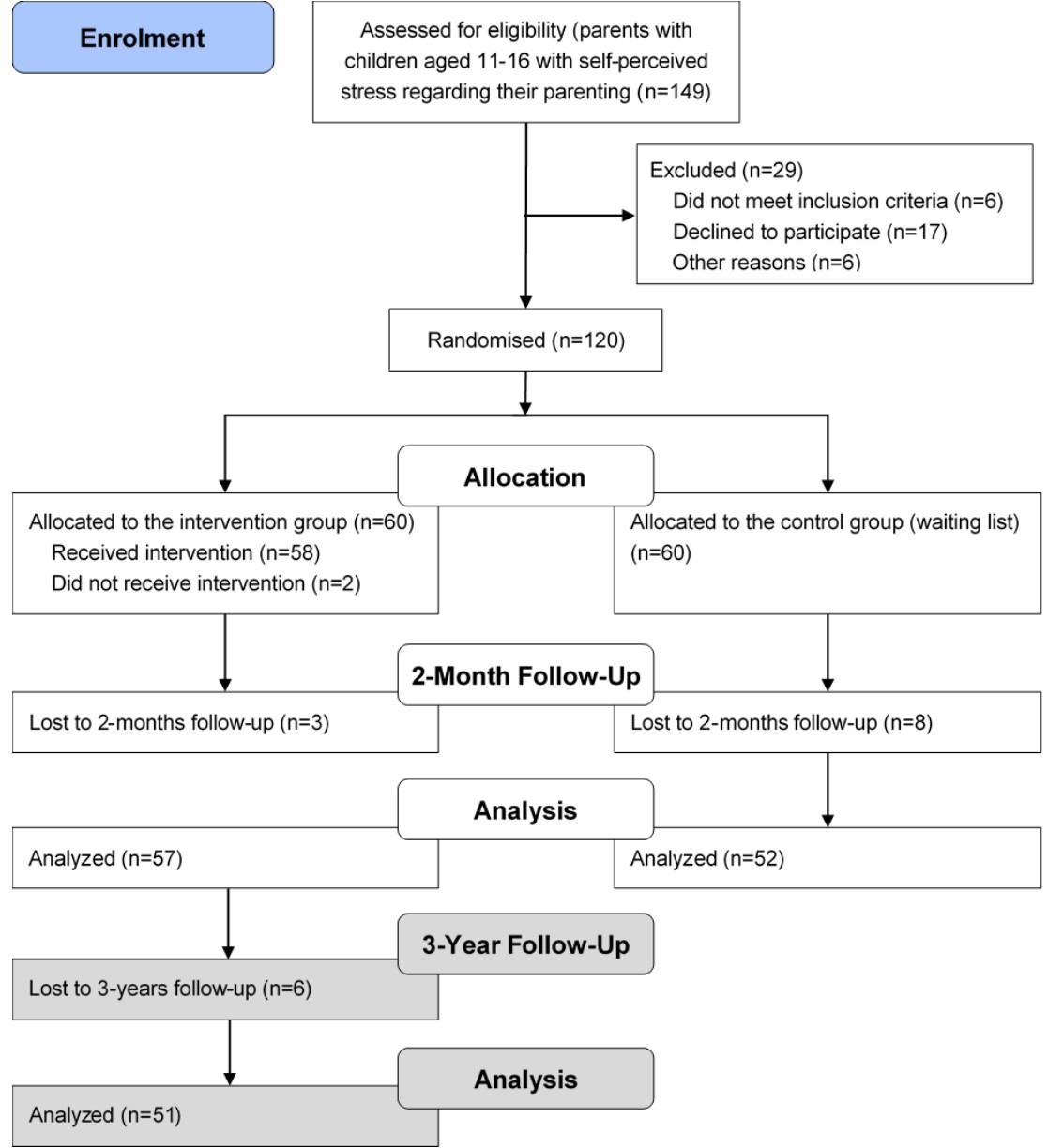

**Figure 1** Participant flow chart throughout the study (grey marked area concerns the current study).

authorities, such as the migration agency, social welfare services and hospitals.

## Outcome measures

The Child Behaviour Checklist (CBCL) for children aged 6–18 years (CBCL 6–18)[30] was used to measure the children's mental health from the parents' perspective. We opted to use the CBCL 6–18 even for the children who had already turned 18 at the 3-year follow-up because: (1) all the children were still living at home with their parents and (2) using the adult version would have led to difficulties in estimating the changes in the level of problems from baseline. The CBCL 6–18 consists of 133 items encompassing two sections: (1) competence scales (20 items on activities and social scales) and (2) emotional and behavioural problems (113 items). In this study, only items related to emotional and behavioural

problems were used. According to the Achenbach System of Empirically Based Assessment (ASEBA) guideline,[31] alcohol- and sex-related items can be excluded for Muslim respondents. In this study, five sex-related items were considered culturally inappropriate for our target population and thus, they were omitted, leaving 107 items in the CBCL. There were three possible response options for each problem item: 0 (not true), 1 (somewhat or sometimes true) and 2 (very true or often true). Using the ASEBA manual for CBCL 6–18,[31] we constructed six variables: a total problem score, two broad groupings of emotional and behavioural problems (internalising and externalising problems) and three syndrome scales (social, thought and attention problems). Higher scores indicate greater problem severity. In the current sample, Cronbach's alphas at baseline, 2-month follow-up and

3-year follow-up were, respectively, 0.82, 0.75 and 0.69 for total problem scores and 0.85, 0.73 and 0.75 for externalising problems. Cronbach's alphas for internalising problems and the three syndrome scales were low (for a detailed report please see online supplemental table 2S) and thus we opted to report the results for them only in online supplemental table 3S and figure 1S.

The 12-item version of the General Health Questionnaire (GHQ-12)[32] was used to measure the parents' overall mental health over the previous few weeks. Each item was rated on a four-point Likert scale from 1 (better than usual) to 4 (much worse than usual), with higher scores indicating higher mental health distress. Cronbach's alphas for GHQ-12 in this study were 0.80 (baseline), 0.91 (2-month follow-up) and 0.81 (3-year follow-up).

Both CBCL 6–18 and GHQ-12 were translated into Somali following the five steps of the WHO's process of translation.[33] The translation started with forward translation (from English to Somali) by the first author and an interpreter who worked independently. After the translations were compared, a reconciled version was agreed on. In the second step, backward translation was conducted by a bilingual social worker. In the third step, the backward translation was compared with the original version and then with the Somali version to check if the forward translation was accurate. In the fourth step, the Somali version of CBCL 6–18 and GHQ-12 were pilot tested with three mothers and fathers for accuracy. In the last step, the revision was proofread and approved by the ASEBA for CBCL 6–18 and by Mapi Research Trust for GHQ-12.

### Statistical methods

All analyses were performed using IBM SPSS Statistics (V.26). There were a few cases of missing data at the item level because some participants failed to answer all the questions. Thus, we started the analysis by reconstructing the mean outcome scores for participants who had at least filled in 70% of the items on each scale or subscale. This resulted in the retention of all 51 cases for CBCL and 49 cases for GHQ-12 in the analyses.

Descriptive statistics of baseline data were presented with mean (SD) or proportions (%). We used a series of t-tests and $\chi^2$ tests to compare the background characteristics of those who remained in the study at the 3-year follow-up and those who dropped out. One-way repeated measure analysis of variance (ANOVAs) were used to compute the within-subject change over time from baseline to 2-month follow-up and 3-year follow-up. We started by performing omnibus tests (whether the three means were different overall). When omnibus tests were significant (p<0.05), we carried out pairwise tests to specifically examine differences from baseline to 2-month follow-up and from baseline to 3-year follow-up. During these post hoc analyses, we adjusted the p values for multiple comparison using Bonferroni method. Cohen's d effect size was calculated from F tests (d=0.2, small; d=0.5, medium and d=0.8, large).[34]

In addition, the clinical significance of change was measured from baseline to 2-month follow-up and 3-year follow-up using the method recommended by Jacobson and Truax.[35] Population norms for the CBCL 6–18 and GHQ-12 were not available for the present study population. Therefore, we used the pretest scores for all participants in the original randomised controlled trial to calculate the SE of differences. We assumed a measurement reliability of 0.8 for each measure. The clinical significance of change was then presented as the proportion of children/parents who had deteriorated, remained unchanged or improved at 2 months and 3 years follow-ups.

## RESULTS

### Attrition and baseline characteristics

Table 1 presents baseline characteristics of the participants who remained in the study at the 3-year follow-up and those who dropped out. No statistically significant differences were found between the two groups. Of the 51 participants who remained at the follow-up and were included in the analyses, 34 were mothers and 17 were fathers. Only five of them were employed at the time. Most of them had not entered upper secondary school, were married and living with their partner. The six participants who dropped out of the study at the 3-year follow-up were all mothers, were educated up to upper secondary school level only, and were unemployed. Five of these six were not cohabiting with a partner.

A majority of the parents (n=40) attended at least eight of the 12 intervention sessions. Most parents also received support from the group leaders in the form of reading letters from government agencies. However, this information was not formally recorded and thus we do not have the exact number of parents who used this service.

### Long-term findings

Although most variables were not normally distributed, ANOVA is considered relatively robust to violations of normality assumptions.[36 37]

Mauchly's test indicated that the assumption of sphericity had been violated for all the three outcomes, therefore, following the recommendations by Howell,[38] we report multivariate tests (Pillai's Trace). The results show significant improvement over time for all outcomes (see table 2 for means, SD, mean differences, F values, and 95% CIs and effect sizes).

For parent-reported CBCL, repeated measure ANOVA revealed significant improvements in children from baseline to the 3-year follow-up for both total problem scores (95% CI 11.49 to 18.00) and externalising problems (95% CI 2.48 to 5.83). The effect sizes were all large. Similarly, significant improvements were observed for parents' mental health as measured by GHQ-12. The associated effect sizes were medium.

### Clinical significance of change in children's and parents' mental health

Figure 2 shows the clinical significance of change in the mental health of children and parents at 2-month

**Table 1** Characteristics of intervention participants at baseline for those who remained in the study (n=51) and those who dropped out after 2-month follow-up (n=6)

| | Remained | | Dropped out | |
|---|---|---|---|---|
| Categorical variables | n | % | n | % |
| Parents | | | | |
| Mothers | 34 | 66.7 | 6 | 100.0 |
| Fathers | 17 | 33.3 | 0 | 0.0 |
| Identified child's gender | | | | |
| Girls | 32 | 62.7 | 2 | 33.3 |
| Boys | 19 | 37.3 | 4 | 66.7 |
| Years in Sweden * | | | | |
| 1–5 years | 20 | 39.2 | 1 | 16.7 |
| 6 years or more | 31 | 60.8 | 5 | 83.3 |
| Highest educational level | | | | |
| Less than upper secondary school | 29 | 56.9 | 6 | 100.0 |
| Upper secondary school or more | 22 | 43.1 | 0 | 0.0 |
| Occupation | | | | |
| Employed | 5 | 9.8 | 0 | 0.0 |
| Other | 46 | 90.2 | 6 | 100.0 |
| Civic status | | | | |
| Married | 33 | 64.7 | 4 | 66.7 |
| Single | 18 | 35.3 | 2 | 33.3 |
| Cohabiting with partner | | | | |
| Yes | 30 | 58.8 | 1 | 16.7 |
| No | 21 | 41.2 | 5 | 83.3 |
| Worries about own financial situation † | | | | |
| Yes | 17 | 34.0 | 3 | 50.0 |
| No | 33 | 66.0 | 3 | 50.0 |
| Continuous variables | M | SD | M | SD |
| Identified child's age at baseline | 13.51 | 1.61 | 14.17 | 2.14 |
| Participants' age | 43.80 | 7.77 | 46.33 | 11.08 |
| Number of children living at home‡ | 5.74 | 2.77 | 4.33 | 2.94 |

No statistically significant differences were found between the two groups.
*None of the parents in our sample had lived in any other Western country before arriving in Sweden. Typically, the journey from Somali to Sweden takes less than a year with some Somalis having to wait in Somali's neighbouring countries before continuing their journey to Europe and resettling in Sweden.
†n=50 for those who remained in the study.
‡n=49 for those who remained in the study.

and 3-year follow-ups. The proportion of children who exhibited clinically significant change from baseline to the 2-month follow-up was 80% for total problem score and 82% for externalising problems. This indicated that the level of problems remained unchanged for most children. Very few children exhibited negative changes (4% and 6%), while positive changes were observed for a higher number of children (16% and 12%).

The positive changes were more pronounced from baseline to the 3-year follow-up: 51% of children demonstrated clinically significant improvement on total problem score and 26% on externalising problems. No negative changes were observed for children in either of the outcomes.

Most parents showed no clinically significant changes from baseline to the 2-month or 3-year follow-ups (75% and 86%, respectively). Very few showed negative changes (6% and 2% at the 2-month and 3-year follow-ups, respectively), while positive changes were observed for a higher number of parents (20% and 12% at the 2 months and 3 years follow-ups, respectively).

Supplementary and sensitivity analyses: Long-term follow-up data were available for 36 out of 39 parents in the control group who had received the intervention. One-way repeated measure ANOVAs on this group revealed similar significant improvement from baseline to the long-term follow-up for all the outcomes. In addition, we imputed the data for the six participants who had dropped out from the study using the worst available scores (ie, worst-case scenario). One-way repeated measure ANOVAs on the imputed data revealed similar significant improvement from baseline to the long-term follow-up for all the outcomes. These results point to the robustness of the main findings.

## DISCUSSION

To the best of our knowledge, this is the first study to evaluate the long-term effect of a culturally tailored parenting support programme on the mental health of immigrant parents and their children. Our findings reveal that all the short-term positive changes in the mental health of children and parents were maintained 3 years postintervention. The findings also suggest that clinically positive changes were generally retained over time.

Our findings are in line with previous systematic and meta-analytical reviews,[39 40] which have indicated that positive change in children's emotional and behavioural problems is sustained over time. However, the parenting programmes evaluated in these previous studies were mainly based on cognitive behavioural theories. In contrast, we tested the Connect programme in our study, which is based on attachment theory. Our findings are also consistent with another Swedish study on non-immigrant parents that showed that the Connect programme had continuously improved children's behavioural and emotional problems 2 years after the intervention,[41] although the related effect sizes were generally larger in our study.

The improvement in children's mental health can be explained by parents' reflectivity and sensitivity towards their children's emotional needs, which is the

Table 2  Within-group comparison of intervention participants at baseline, 2-month follow-up, and 3-year follow-up (N=51)

| | | | | | Within-subject change | | | | | | | | |
| | | | | | Baseline to 2 months FU | | | | Baseline to 3 years FU | | | | |
| | Baseline mean (SD) | 2 months FU mean (SD) | 3 years FU mean (SD) | F (2, 49)† | Mean diff | F (1, 50)‡ | 95% CI | d | Mean diff | F (1, 50)‡ | 95% CI | d |
|---|---|---|---|---|---|---|---|---|---|---|---|---|
| **CBCL 6–18** | | | | | | | | | | | | |
| Total problem score | 15.71 (9.8) | 9.49 (7.4) | 0.96 (2.1) | 95.86*** | 6.22 | 15.13*** | 2.26 to 10.18 | 0.54 | 14.75 | 125.84*** | 11.49 to 18.00 | 1.57 |
| Externalising problems | 4.67 (5.2) | 2.39 (3.5) | 0.51 (1.4) | 26.94*** | 2.28 | 8.28* | 0.32 to 4.23 | 0.40 | 4.12 | 37.77*** | 2.48 to 5.83 | 0.86 |
| GHQ-12 | 20.00 (4.2) | 17.80 (4.8) | 18.22 (2.3) | 5.46** | 2.18 | 7.37* | 0.19 to 4.18 | 0.41 | 1.76 | 10.28** | 0.40 to 3.11 | 0.46 |

*P<0.05, **p<0.01, ***p<0.001.
†F(2, 47) for GHQ-12.
‡P values adjusted for multiple comparison (Bonferroni); F(1, 48) for GHQ-12.
CBCL 6–18, Child Behaviour Checklist for children 6–18 years; FU, follow-up; GHQ, General Health Questionnaire.

aim of the Connect programme.[42] Our previous qualitative study[43] revealed that after participating in the Connect programme, parents became more aware of their children's expressed and non-expressed needs and made themselves more emotionally available to them. Parents also reported using the principles in the Connect programme, such as 'taking a step back'.[43] The positive results can also be attributed to factors unrelated to the intervention. For example, as children grow older, teenage problems and parent-adolescent conflicts diminish, meaning parents report fewer problems.

In this study, a smaller change was detected in parents' mental health. Although parenting support programmes do not aim to improve parents' mental health directly, studies have shown that parents can benefit from parenting support programmes that are tailored to their needs.[44–46] However, other external factors (such as employment, acculturation, perceived discrimination and social isolation) might play a more important role in parental mental health.[7 12 13 47 48] Therefore, the benefits associated with participation in parenting programmes might be more difficult to maintain over time.

### Strengths and limitations
Several strengths and limitations of this study should be taken into consideration. First, there are few studies on the long-term effect of parenting programmes, particularly those delivered to immigrant parents.[49] Second, most studies with long-term follow-ups have high-drop-out rates. In our study, we were successful in retaining 85% of our original sample.

The main limitation of the study was the lack of a control group, which only allowed within-group analyses. The control group was offered the intervention shortly after the 2-month follow-up due to ethical and practical reason. The collaborator municipality was resolved that

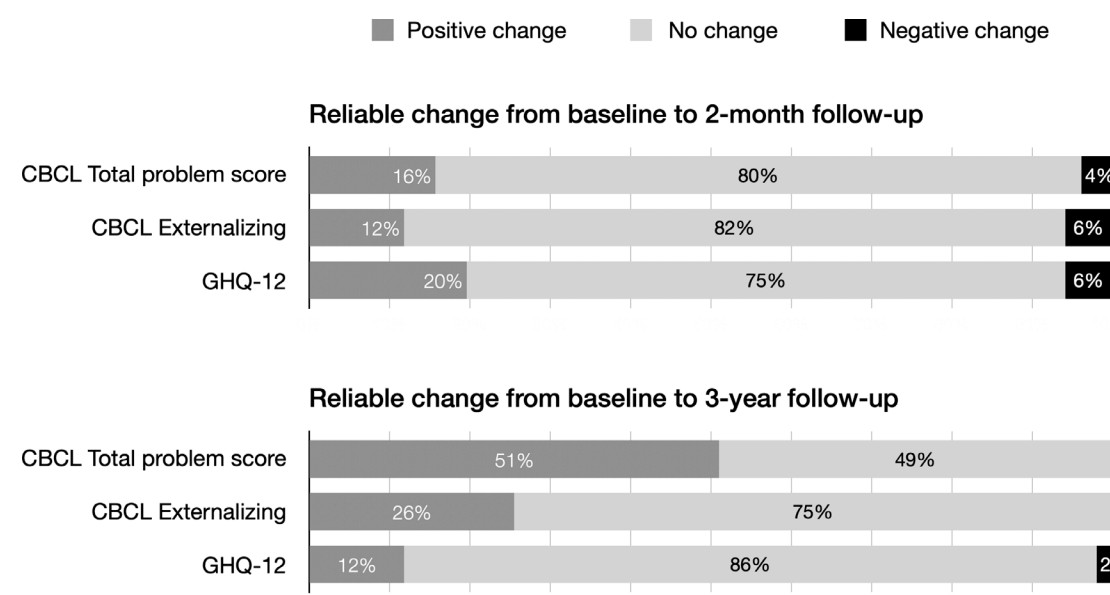

CBCL = Child Behaviour Checklist, GHQ = General Health Questionnaire

Figure 2  Clinical significance of change from baseline: proportion of children (CBCL) and parents (GHQ) showing reliable change (n=51 for children and 49 parents).

the intervention should be given to the parents in the control group because of its short-term positive impact on participating families.

Another limitation is that the outcome measures used in this study were not previously tested for their reliability and validity on this specific population. In the current sample, internal consistency was low for some of the CBCL subscales. However, both CBCL and GHQ are well-established questionnaires that have been readily used in cross-cultural settings showing good reliability and validity.[32 50] Thus, it is important to examine Somali parents understanding of the individual items and also test the reliability and validity of these two measures, particularly CBCL, in a larger sample of Somali parents.

Finally, similar to many other studies, the intervention was evaluated using parental reports only, which may be biased. It is preferable to collect data from parents, children and teachers. In our study, parents might have had reservations about reporting problems, as highlighted in previous studies.[7 22] We found that parents reported a very low number of problems at the 3-year follow-up. However, this can be normative for children at this age, or it might indicate that attending the programme changed parental perceptions of their children's behaviour. Including children's self-reports would have strengthened the results. We had, in fact, initially intended to evaluate the intervention using both parents' and children's reports. However, the vast majority of the participating parents refused to consent to their children's direct participation in the study. This was mainly due to their (mis)perception of child protection policies in Sweden; they were afraid that the children might, accidentally or intentionally, say something true or untrue that would lead to the involvement of Family and Child Welfare Services and their children would potentially be taken away. Interestingly, at the 3 years follow-up, parents readily agreed with their children participation in a focus group discussion. This implies that addressing misconceptions about child protection policies and building a long-term relationship with immigrant communities might facilitate the issues regarding children's participation in research studies.

## CONCLUSION AND IMPLICATION FOR CLINICAL PRACTICE

Previous studies have shown that participation in a culturally tailored parenting programme was associated with positive changes in the mental health of Somaliborn parents and their children. In the current study, we showed that these positive changes were maintained 3 years postintervention. Given the study's limitations, our findings need to be replicated. Future studies should strive to measure the long-term effects of parenting programmes for immigrant parents using a control group, and by including well-validated ratings from parents, children and teachers. Despite the reported limitations, the study highlights the potential benefits of offering a culturally tailored parenting support programme to immigrant families.

**Acknowledgements** The authors would like to thank all the parents who participated in this study. We also thank research assistant Abdikerim Mohamed for his important work in the data collection, and the Kamprad Foundation for funding this study.

**Contributors** All authors conceptualised and designed the study. FO, LV and RS were responsible for data analyses and interpretation. They drafted the initial manuscript, reviewed, revised and approved the final manuscript as submitted. RF reviewed the data analyses and interpretation. She revised, reviewed and approved the final manuscript as submitted. U-KS reviewed, revised and approved the final manuscript as submitted. MK-A reviewed, revised and approved the final manuscript as submitted. All authors approved the final manuscript as submitted.

**Funding** This work was supported by the Kamprad Foundation, grant number 2017:0069.

**Competing interests** None declared.

**Patient consent for publication** Not required.

**Ethics approval** The Swedish Regional Ethical Review Board in Uppsala, Sweden (Dnr 2014/048/2).

**Provenance and peer review** Not commissioned; externally peer reviewed.

**Data availability statement** Data are available on reasonable request. The data sets for this study is not open access, but can be made available upon request to the Principal Investigator (FO) and according to the ethical approval.

**ORCID iD**
Fatumo Osman http://orcid.org/0000-0002-0038-9402

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
