## [Reviewer comments · BMJ Open]

ARTICLE DETAILS

TITLE (PROVISIONAL)	The impact of a culturally tailored parenting program on the mental health of Somali parents and children living in Sweden: A longitudinal cohort study
AUTHORS	Osman, Fatumo; Vixner, Linda; Flacking, Renee; klingberg-allvin, marie; Schön, Ulla-Karin; Salari, Raziye

VERSION 1 – REVIEW

REVIEWER	Thompson, MJJ University of Southampton
REVIEW RETURNED	11-Nov-2020

GENERAL COMMENTS	This is the report of an intervention study providing parenting support to a population of Somali immigrants to Sweden. The authors highlight the stress factors for this population ICK of social networks and support, perceived immigration, inadequate accommodation and a cultural challenges. Also possible low confidence in parenting and the use of more authoritarian parenting It is postulated that parenting programmes that are culturally tailored to a group of immigrant parents needs would improve parenting skills, improve parental mental health and children's. it could also enhance parent child relationships. The authors wanted to see if such a programme would continue to show positive outcomes in the measures at a three year follow up. Unfortunately for the validity of the follow up results as the control group was offered the intervention the control group were lost to follow up This therefore was a within group comparison. Intervention The parenting intervention was the Connect parenting parent support training. It was adapted culturally to this group of parents with the programme delivered in the participants' native language using group leaders of similar experiences to the parents and using role play and case examples The societal information was delivered by a Swedish speaking professional translated by a group leader Primary outcome: CBCL total, internalising and externalising problems Secondary outcomes: CBCL on symptoms of social, thought and attention problems and GHQ-12 scores
--

	Some items of the CBCL were removed for Muslim respondents (alcohol and sex related questions). Parents filed this in. Therefore no blinded measures. ANOVAs were used to measure within subject change Change at two months and Three year FU. Pretrial subject change of all subjects in the initial trial was used to calculate the standard error of difference. Clinical significance of change was the proportion of children/parents who had deteriorated, remained the same for improved at two month and at three year follow up. No difference in group that stayed in or dropped out Statistical methods used appropriate. Results Show improvement in all measures with large effects sizes in primary measures and medium to large in the secondary measures Clinical significance of change in the mental health of children and parents was better at the 3 year follow up compared with the two month follow up. Parents did not show clinically significant change, with no negative outcomes. Only 12 % to 20% showed positive changes. Conclusion A well written paper with a well planned study. The authors have taken into account the problems facing immigrant families and have tried to address them in the trial with a parenting group adapted to the cultural needs of the research population but also offering support with housing, finance and coping with a different culture. From reading other papers from this group I think they also offer individual support as well. If this is to be replicated it would be useful to know the number of attendances at the group and individual contact. As with other programmes that also makes a difference but is seldom documented (e.g. IY). The authors highlight that this is a within group evaluation with no blinded measures of change. However an interesting study and important to work with different cultures and highlight this.
--	--

REVIEWER	Tan, Kit Aun Universiti Putra Malaysia Faculty of Medicine and Health Sciences, Department of Psychiatry
REVIEW RETURNED	06-Jan-2021

GENERAL COMMENTS	Thank you for the opportunity to review this manuscript. The authors have done a good job on building up a case for support, arguing that a longitudinal study to examine the effectiveness of a
--

culturally tailored parenting support program on improving Somali parents' and their children's mental health is needed. It is difficult to assess the overall quality of this manuscript because of the inadequate descriptions on the Methods (e.g., sequence generation, allocation concealment mechanism, binding, etc.) as per the CONSORT Statement. Here are my comments/feedback for the Editor's and authors' consideration.

1. I refer to the Abstract.

a. For reporting of randomised controlled trials, please use the appropriate extension to the CONSORT statement, including the extension for writing abstracts.

2. I refer to the Introduction.

a. An important piece missing from the Introduction is discussions on internalizing problems and externalizing problems as primary outcome variables and on social, thought, and attentional problems, and general mental health as secondary outcome variables. The authors attempted some discussions but I feel this could still be better justified by building up a case for theoretical background, arguing that a culturally tailored parenting support program—that is the Ladnaan intervention—could be delivered to reduce/increase study outcomes.

3. I refer to the Methods.

a. Another issue of concern I have about this study is its lack of research transparency—reports of randomized trials must conform to the CONSORT Statement. It is not a good idea to refer to another study when describing a trial design and methods (sample size, randomization, & implementation). Brief information for the trial design and methods should be included in the current study, so readers do not have to search other articles.

b. Sample specific Cronbach alpha values need to be provided for all scale scores used in the study. The authors need to present at minimum, their own present study's Cronbach alphas for that particular scale or scales used. Therefore, the present sample's Cronbach alpha should be presented for the Somali versions of the Child Behavior Checklist (CBCL) and the General Health Questionnaire. Further, if the subscales of the CBCL were used, then their Cronbach alphas should be reported. This permits future meta-analytic work.

c. The instrument translation process is sparsely described and could use more detail in terms of forward translation, expert panel, backward translation, etc.

4. I refer to the Results.

a. Evidence for statistical assumptions should be provided. At the least, the authors need to present evidence for normality assumption (e.g., skewness and kurtosis indices).

b. My next comment touches on the treatment of missing data. It is not clear how the authors dealt with missing data. For multivariate analyses, the study's dataset was only limited to 51 participants from the intervention group. Please check if per-protocol analyses were performed.

	c. The authors used the worst-case scenario to impute missing data. I feel a greater and proper emphasis on this particular imputation method is needed. 5. I refer to the Discussion. a. The authors could have considered issues surrounding the reliability of study measures. The study used two outcome measures, there may have been many extraneous variables that were not measured which could have influenced the results. 6. Other issues. a. Minor typographical errors*. i. Use italics for statistical symbols (other than vectors and matrices): n, F, M, p, SD, etc. * Read through the manuscript again to catch these similar minor typographical errors. +++The End+++
--	---

VERSION 1 – AUTHOR RESPONSE

Reviewer 1, comment: A well written paper with a well planned study. The authors have taken into account the problems facing immigrant families and have tried to address them in the trial with a parenting group adapted to the cultural needs of the research population but also offering support with housing, finance and coping with a different culture. From reading other papers from this group I think they also offer individual support as well. If this is to be replicated it would be useful to know the number of attendances at the group and individual contact. As with other programmes that also makes a difference but is seldom documented (e.g. IY).	Thank you, we are pleased to hear that the paper is interesting and well-written. Regarding the number of attendees of group sessions and the number of participants who have received individual support has been added in the results section. See page 11 The text states now: The majority of parents (n=40) attended at least 8 of the 12 intervention sessions. Most parents also received support from the group leaders in the form of reading letters from government agencies. However, this information was not formally recorded.
---	---

Reviewer 2, comment #1: 1. I refer to the Abstract. a. For reporting of randomised controlled trials, please use the appropriate extension to the CONSORT statement, including the extension for writing abstracts.	Since this is longitudinal cohort study, we have therefore provided a completed STROBE checklist according to the BMJ Open guidelines.
Reviewer 2, comment #2: I refer to the Introduction. a. An important piece missing from the Introduction is discussions on internalizing problems and externalizing problems as primary outcome variables and on social, thought, and attentional problems, and general mental health as secondary outcome variables. The authors attempted some discussions but I feel this could still be better justified by building up a case for theoretical background, arguing that a culturally tailored parenting support program—that is the Ladnaan intervention—could be delivered to reduce/increase study outcomes.	We have now elaborated more on the impact of migration on children and parents' mental health. We have also mentioned that there is already a large body of research supporting the effectiveness of parenting support programs for improving the mental health of children and their parents. However, these programs need to be culturally sensitive. The new text is on page 4.
Reviewer 2, comment #3a: I refer to the Methods. a. Another issue of concern I have about this study is its lack of research transparency—reports of randomized trials must conform to the CONSORT Statement. It is not a good idea to refer to another study when describing a trial design and methods (sample size, randomization, & implementation). Brief information for the trial design and methods should be included in the current study, so readers do not have to search other articles.	Since this is longitudinal cohort study, we have therefore provided a completed STROBE checklist according to the BMJ Open guidelines. We have now tried to make it clearer that the current study is a longitudinal cohort study and not a randomized control trial (e.g., see the revised title and abstract please). Because the current study focuses on the follow up of the original intervention group ONLY, we believe detailed information about randomization process is not of high relevance here and rather than adding clarity might contribute to more confusion about the study design. If the reviewer/editor disagrees with us, we can, of course, add this information.

Reviewer 2, comment #3b: b. Sample specific Cronbach alpha values need to be provided for all scale scores used in the study. The authors need to present at minimum, their own present study's Cronbach alphas for that particular scale or scales used. Therefore, the present sample's Cronbach alpha should be presented for the Somali versions of the Child Behavior Checklist (CBCL) and the General Health Questionnaire. Further, if the subscales of the CBCL were used, then their Cronbach alphas should be reported. This permits future meta-analytic work.	We have added the Cronbach alphas for CBCL and GHQ-12 used in this study, please see page 8.
Reviewer 2, comment #3c: c. The instrument translation process is sparsely described and could use more detail in terms of forward translation, expert panel, backward translation, etc.	We have described the five steps we applied for the translations of the CBCL 6-18 and GHQ-12. Please find in page 9.
Reviewer 2, comment #4a: I refer to the Results. a. Evidence for statistical assumptions should be provided. At the least, the authors need to present evidence for normality assumption (e.g., skewness and kurtosis indices).	We had previously stated that assumption of sphericity was tested. Now, we have also added a sentence about the normality assumption: Although most variables were not normally distributed, ANOVA is considered relatively robust to violations of normality assumptions [36, 37]. Please see page 11
Reviewer 2, comment #4b: My next comment touches on the treatment of missing data. It is not clear how the authors dealt with missing data. For multivariate analyses, the study's dataset was only limited to 51 participants from the intervention group. Please check if per-protocol analyses were performed.	As mentioned in page 7, out of 60 families randomized to the intervention group, 3 were lost to the 2-month follow-up and another 6 to the 3-year follow-up. We have now added some details about the intervention exposure (please see our response to the first reviewer's comment). We performed the main analyses on all 51 cases who participated in the 3-year follow-up (see page 11). As the reviewer notes in his next comment, in a series of sensitivity analyses, we used worse-case scenario to

	impute missing data for the 6 participants lost to the 3-year follow-up. We have also added information on how we dealt with missing data at item level, please see page 9
Reviewer 2, comment #4c: The authors used the worst-case scenario to impute missing data. I feel a greater and proper emphasis on this particular imputation method is needed.	Although multiple imputation is widely considered the preferred method for data imputation, it is not a viable option for analyses of variance (such as repeated ANOVAs) because as far as we know currently there is no method available for pooling the results for analyses of variance. In addition, the low number of participants lost to follow up means missing data is not a major issue in this study. To keep the manuscript succinct, we have refrained from spelling these points out. If the reviewer/editor believes this information will be of high value for readers, we can of course add them to the results section.
Reviewer 2, comment #5: I refer to the Discussion. a. The authors could have considered issues surrounding the reliability of study measures. The study used two outcome measures, there may have been many extraneous variables that were not measured which could have influenced the results.	We have discussed the strength and limitation of the measures. Please see 15

Reviewer 2, comment #6: Other issues. a. Minor typographical errors. Read through the manuscript again to catch these similar minor typographical errors. i. Use italics for statistical symbols (other than vectors and matrices): n, F, M, p, SD, etc. *	We have now corrected the typographical errors such as using italics for statistical symbols (not marked).
--	---

VERSION 2 – REVIEW

REVIEWER	Thompson, MJJ University of Southampton
REVIEW RETURNED	26-Mar-2021

GENERAL COMMENTS	The authors have addressed the concerns in this revision. The main limitations which they have addressed is the lack of a control group. The other limitation is that the families got a lot of support as well as the parenting programme which would be hard to replicate without a manual but I think that is what they have.
--

REVIEWER	Tan, Kit Aun Universiti Putra Malaysia Faculty of Medicine and Health Sciences, Department of Psychiatry
REVIEW RETURNED	10-Mar-2021

GENERAL COMMENTS	Thank you for the opportunity to review this manuscript. Here are my comments/feedback for the Editor's and authors' consideration:  1. The authors were responsive to comments brought up during the first review. Their revisions were thoughtful. However, there are other minor concerns that I have which will be outlined below. 2. I refer to the Methods.  a. It is insufficient to state that the Cronbach alpha values for the Child Behavior Checklist (CBCL) subscales were above .78. Specific Cronbach alpha values need to be provided for all CBCL subscales (Total Problem, Internalizing Problems, Externalizing Problems, Social Problems, Thought Problem, & Attention problems) for research transparency. b. The authors did not mention Bonferroni adjustment as part of their data analytic plan. It was only in Table 2 when readers were told that the authors adjusted p values for family-wise error rate did readers come to realize the computation of the Bonferroni adjustment. This information needs to be stated much earlier.
---

	3. I refer to Table 2. a. The authors mentioned that no statistical significant differences in sociodemographic variables were found between those who remained in the study and those who dropped out after two-month follow-up. How was this determined? b. Please check if the information on degree of freedom is accurate. Perhaps, the authors could explore a better way to present such information. 4. Other issues. a. Minor typographical errors*. i. Please see Table 2: Please define what “6–18” is. “Internalizing” should be “Internalizing Problem” and “Externalizing” should be “Externalizing Problem”. ii. Please see Supplementary Table 1, Session 1: “child’s developmental” should be “child development”. * Read through the manuscript again to catch these similar minor typographical errors. +++The End+++
--	---

VERSION 2 – AUTHOR RESPONSE

Editor/Reviewer Comments	Author’s Response
--------------------------

Reviewer 2, comment #1: It is insufficient to state that the Cronbach alpha values for the Child Behavior Checklist (CBCL) subscales were above .78. Specific Cronbach alpha values need to be provided for all CBCL subscales (Total Problem, Internalizing Problems, Externalizing Problems, Social Problems, Thought Problem, & Attention problems) for research transparency.

Thank you for insisting that we should provide individual alphas. We recalculated all alphas and realized that internal consistency of CBCL internalizing problems and CBCL syndrome scales were low. Thus, the results for these outcomes were removed from the main paper and presented in a supplementary table only.

“In the current sample, the α Cronbach coefficients at baseline, two-month follow-up and three-year follow-up were respectively .82, .75, and .69 for total problem scores and .85, .73, and .75 for externalizing problems. The α Cronbach coefficients for internalizing problems and the three syndrome scales were low (for a detailed report please see Supplementary Table 2S) and thus we opted to remove the analyses for these outcomes from the main paper and present them only in Supplementary Table 3S.”

“The α Cronbach coefficients for GHQ-12 in this study were .80 (baseline), .91 (two-month follow-up) and .81 (three-year follow-up).”

We have added this as a study limitation:

“Another limitation is that the outcome measures used in this study were not previously tested for their reliability and validity on this specific population. In the current sample, internal consistency was low for some of the CBCL subscales. However, both CBCL and GHQ are well-established questionnaires that have been readily used in cross-cultural settings showing good reliability and validity [32, 50]. Thus, it is important to examine Somali parents understanding of the individual items and also test the reliability and validity of these two measures, particularly CBCL, in a larger sample of Somali parents.”

--	--

Reviewer 2, comment #2: The authors did not mention Bonferroni adjustment as part of their data analytic plan. It was only in Table 2 when readers were told that the authors adjusted p values for family-wise error rate did readers come to realize the computation of the Bonferroni adjustment. This information needs to be stated much earlier.	We have now added this information to the section called “statistical methods”: “We started by performing omnibus tests (whether the three means were different overall). When omnibus tests were significant ($p < .05$), we carried out pairwise tests to specifically examine differences from baseline to two-month follow-up and from baseline to three-year follow-up. During these post-hoc analyses, we adjusted the p values for multiple comparison using the Bonferroni method.”
Reviewer 2, comment #3: I refer to Table 2.  1. The authors mentioned that no statistical significant differences in sociodemographic variables were found between those who remained in the study and those who dropped out after two-month follow-up. How was this determined? 2. Please check if the information on degree of freedom is accurate. Perhaps, the authors could explore a better way to present such information. 	 1. We have now specified how the differences were determined: “We used a series of t-tests and chi-square tests to compare the background characteristics of those who remained in the study at the three-year follow-up and those who dropped out.” Page xx 2. We have presented the degree of freedom using the standard method. The first number in parenthesis refers to the degree of freedom for number of groups and the second number refers to the degree of freedom for number of participants.

Reviewer 2, comment #4: Other issues. a. Minor typographical errors*. i. Please see Table 2: Please define what “6–18” is. “Internalizing” should be “Internalizing Problem” and “Externalizing” should be “Externalizing Problem”. ii. Please see Supplementary Table 1, Session 1: “child’s developmental” should be “child development”.	We have now corrected the typographical errors.
--	--

VERSION 3 – REVIEW

REVIEWER	Tan, Kit Aun Universiti Putra Malaysia Faculty of Medicine and Health Sciences, Department of Psychiatry
REVIEW RETURNED	08-Jul-2021
GENERAL COMMENTS	Thank you for the opportunity to review this manuscript. The authors were responsive to comments brought up during the second review. Their revisions were thoughtful.